# iLLaVA: An Image is Worth Fewer Than $\frac{1}{3}$ Input Tokens in Large Multimodal Models

**Lianyu Hu[1], Liqing Gao[2], Fanhua Shang[1], Liang Wan[1,3]✉*, Wei Feng[1,3],**

[1]School of Computer Science and Technology, Tianjin University,
[2]School of Computer Science and Technology, Tiangong University,
[3]Key Research Center for Surface Monitoring and Analysis of Relics, State Administration of Cultural Heritage
{hly2021}@tju.edu.cn, {lqgao}@tiangong.edu.cn, {fhshang, lwan, wfeng}@tju.edu.cn,
 Code: https://github.com/hulianyuyy/iLLaVA

## Abstract

Recent methods have made notable progress in accelerating Large Vision-Language Models (LVLMs) by exploiting the inherent redundancy in visual inputs. Most existing approaches, however, focus narrowly on reducing image tokens before or within the Large Language Model (LLM) stage to lower computational cost. This overlooks other major bottlenecks, particularly the image encoder, which itself requires substantial computation. As a result, these methods fall short of achieving true end-to-end acceleration. Importantly, the image encoder is the primary contributor of input tokens to the LLM. Thus, reducing visual redundancy at the encoder stage not only speeds up the encoder itself but also significantly lightens the workload for the subsequent LLM. Motivated by this, we investigate how to jointly optimize the image encoder and the LLM along with other LVLM components for comprehensive acceleration. To mitigate the risk of performance degradation from token reduction, we propose a novel token merging strategy that recycles useful information from otherwise discarded tokens. Our approach, iLLaVA, delivers consistent improvements across both image and video understanding tasks, achieving up to a $2\times$ throughput boost and a $4\times$ reduction in prefilling time. Notably, iLLaVA enables a larger model (e.g., InternVL-2.5 26B) to surpass a smaller counterpart (e.g., InternVL-2.5 8B) in both accuracy and efficiency. Extensive comparisons with state-of-the-art token pruning and merging techniques demonstrate the clear superiority of our method. Finally, we provide detailed visualizations for the merging steps of iLLaVA , offering deeper insights into how different LVLM components contribute to efficient computation.

## 1 Introduction

Over the past several years, Large Vision-Language Models (LVLMs) Li et al. (2024a); Bai et al. (2025); Chen et al. (2024c) have demonstrated tremendous progress and enabled promising applications in various downstream tasks including cross-model retrieval Alayrac et al. (2022); Li et al. (2023), medical image analysis Li et al. (2024a); Saab et al. (2024), and video understanding Maaz et al. (2023); Wang et al. (2024c). Within these impressive models, images are usually first divided into patches and then transformed into token sequences, finally fed into a Large Language Model (LLM) and concatenated with text instructions to generate the desired textual outputs. The direct combination of image tokens and textual contexts already shows impressive performance over a broad range of tasks, which enables the emergence of powerful AI tools like GPT-4o, Gemini 1.5 pro, DALL·E3 and Kling.

Despite their remarkable success, LVLMs face severe challenges in terms of computational complexity and resource demands. These models typically rely on self/cross-attention mechanisms to capture input information. However, the inherent $\mathcal{O}(n^2)$ complexity of attention operations makes computation scale quadratically with the number of tokens. For image and video inputs, which can easily expand into thousands or even tens of thousands of tokens, this burden becomes especially

---

*Liang Wan is the Corresponding Author

prohibitive. In practice, running inference with a 34B-parameter LVLM can require around 80GB of memory Xu et al. (2024), far exceeding the capacity of many institutions with limited computing infrastructure. Moreover, the slow inference speed of LVLMs raises serious concerns for applications demanding real-time or near-real-time responses Chen et al. (2024a); Yang et al. (2025). Therefore, improving the inference efficiency of LVLMs is not only a technical necessity but also a critical step for enabling their widespread deployment in real-world scenarios.

Several works Chen et al. (2024a); Xing et al. (2025); Yang et al. (2025); Zhang et al. (2025a); Alvar et al. (2025) have explored reducing image tokens for inference acceleration, motivated by the observation that visual features often contain substantial redundancy. While these approaches have achieved notable progress, they narrowly focus on pruning or compressing tokens before or within the LLM stage to reduce its computational overhead. However, this overlooks another critical bottleneck: the image encoder. As we demonstrate in this paper, the image encoder constitutes a non-negligible portion of the overall computation in LVLMs, and its cost cannot be ignored. More importantly, the encoder is also the largest contributor of tokens to the LLM. Thus, reducing redundancy at the encoder stage not only accelerates the encoder itself but also yields compounded efficiency gains by substantially reducing the input load for the subsequent LLM.

To enable comprehensive acceleration of LVLMs, we propose iLLaVA, which jointly optimizes the image encoder and the LLM which constitute two of the most computationally demanding components in LVLMs. Our design is built on two key innovations. First, unlike prior methods that only reduce tokens within the LLM, iLLaVA performs token reduction in both the image encoder and the LLM. This dual-stage reduction substantially decreases the overall computational budget of the entire model. Second, to mitigate the risk of performance degradation from token reduction, we introduce a novel token merging strategy that recycles potentially informative content from discarded tokens, thereby preserving critical input information. Experiments upon over 10+ image and video understanding tasks show that iLLaVA successfully maintains >95% performance over various token reduction ratios and achieves significantly better results compared to state-of-the-art methods. Furthermore, iLLaVA could $2\times$ the throughput and reduce the prefilling time by $4\times$, and enables a larger model (e.g., , InternVL-2.5 26B) to outperform a smaller model (e.g., InternVL-2.5 8B) with both better performance and higher throughput. Comparison of iLLaVA with recent competitive approaches demonstrates the superiority of iLLaVA over computing efficiency and model accuracy.

## 2 RELATED WORK

### 2.1 LARGE VISION-LANGUAGE MODELS

Our work is closely related to the surge of LVLMs. Traditional methods Ramachandram & Taylor (2017); Xu et al. (2023b); Ochoa et al. (2017) usually collected large vision-language datasets and learned joint representations between vision and language from scratch to handle different tasks. These methods usually worked well in in-domain data but performed inferiorly in out-domain data that requires common sense and world knowledge.

Later, powered by the abundance of high-text data, LVLMs Li et al. (2024a); Wang et al. (2024b); Bai et al. (2025); Chen et al. (2024c) shows impressive performance across a wide range of image understanding Yue et al. (2024a); Liu et al. (2024); Masry et al. (2022); Mathew et al. (2022; 2021) and reasoning tasks Li et al. (2024b); Chen et al. (2024b). These methods usually first transform the input images as patches, and then feed them into a vision-transformer-based image encoder. The extracted features are sent into a projector for dimension projection, whose outputs are further concatenated with the system prompts and user instructions to serve as inputs for the language model to generate textual outputs. As the attention mechanism with computational complexity $\mathcal{O}(n^2)$ is both used in the image encoder and language model in LVLMs, they have to consume high computational resources and own high inference latency when faced with long input sequences. Our method is deployed upon the state-of-the-art Qwen2.5-VL Bai et al. (2025), InternVL-2.5 Chen et al. (2024c), Minicpm-v2.6 Yao et al. (2024) and LLaVA-Onevision Li et al. (2024a) series to accelerate models.

### 2.2 TOKEN REDUCTION

Token reduction has been widely explored in both computer vision Meng et al. (2022); Chen et al. (2023); Pan et al. (2022); Ryoo et al. (2021); Rao et al. (2021) and natural language processing

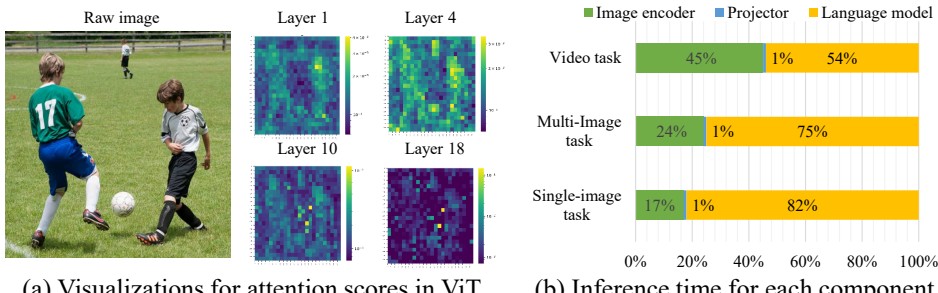

(a) Visualizations for attention scores in ViT     (b) Inference time for each component

Figure 1: (a) Visualizations for the attention scores of different layers in the image encoder. (b) Proportion of inference time for each component in LVLMs across different tasks.

(NLP) Goyal et al. (2020); Kim & Cho (2020); Lassance et al. (2021). However, these methods usually require training, while our method can be done in a training-free manner. In multimodal learning, a series of methods tried to prune the tokens of intermediate layers for model acceleration. FastV Chen et al. (2024a) first proposes to select the TopK-activated tokens within the language model to accelerate the forward pass in a pioneering manner. SparseVLM Zhang et al. (2025b) proposes to select relevant text tokens to rate the significance of visual tokens using self-attention matrices, and then prunes visual tokens using the proposed strategy to maximize sparsity. Faster-VLM Zhang et al. (2025a) utilizes the attention scores between the [CLS] token and image tokens from the visual encoder to conduct token pruning. VisionZip Yang et al. (2025) selects a set of informative tokens before the language model as inputs and merges beneficial information from the discarded tokens. PyramidDrop Xing et al. (2025) partitions the LVLM into several stages and drops part of the image tokens at the end of each stage with a pre-defined ratio. DivPrune Alvar et al. (2025) first formulates token pruning as Max-Min Diversity Problem (MMDP) and then solves the MMDP to obtain the selected subset and prunes the rest. AdaFV Han et al. (2025) introduces a self-adaptive cross-modality attention mixture mechanism to dynamically leverage visual saliency and text-to-image similarity to select informative visual tokens. AIM Zhong et al. (2024) first conducts token merging before the LLM to decrease the input length, and then gradually performs token pruning for the intermediate LLM layers to reduce image tokens. Despite considerable progress on model efficiency, these methods typically just reduce image tokens within or before the LLM, and overlook other computationally-heavy components in LVLMs. In this work, we go beyond this limitation by jointly accelerating both the image encoder and the LLM, enabling more comprehensive and effective efficiency improvements.

## 3 ɪLLᴀVA

### 3.1 ɪɴsɪɢʜᴛs

Recent LVLMs Li et al. (2024a); Wang et al. (2024b); Bai et al. (2025); Chen et al. (2024c) are always consisted of an image encoder to encode input images, whose output features are transformed by a projector and are further combined with textual tokens as inputs into a LLM for processing. Previous efficient methods Chen et al. (2024a); Xing et al. (2025); Yang et al. (2025); Zhang et al. (2025a); Alvar et al. (2025) have made considerable progress in accelerating LVLMs, which usually try to reduce image tokens during or before the LLM to accelerate LVLMs. However, the massive redundancy in other network stages (e.g., image encoder) is overlooked by recent methods, which fail to fully leverage the potential of inherent feature redundancy to accelerate LVLMs. To validate the feature redundancy within the image encoder, we first visualize a raw image and four spatial attention maps from different intermediate layers of the image encoder in Fig. 1(a). One can see that the encoder only pays major attention to a small part of areas in the image (the bird) and overlooks others, which demonstrates that there exists considerable visual redundancy in the image encoder. We also observe that visual redundancy spreads unevenly across different layers, as the distribution of spatial attention maps varies across different layers. In Fig. 1(b), we further plot the inference time distribution for components in LVLMs including the image encoder, projector and LLM. It's observed that the image encoder and the LLM dominate (>99%) the inference time in LVLMs and a large part of the inference time is consumed by the image encoder. The inference time proportion of the image encoder consistently increases as task inputs vary from single-image, multi-image

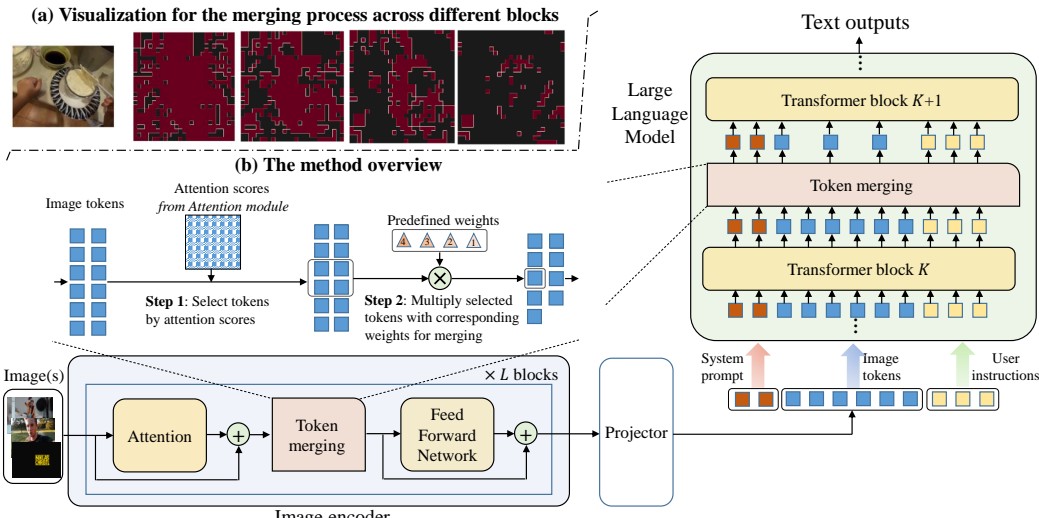

Figure 2: (a) Visualizations for the token merging process across different blocks in the image encoder. The red region denotes selected tokens and the black regions represent discarded tokens. (b) The framework overview for iLLaVA. We apply token merging at the intermediate layers of both the image encoder and the language model to accelerate the forward pass.

to videos, owning up to 45%. Besides, as the image encoder provides quite a large quantity of input tokens for the subsequent LLM, reducing tokens in the image encoder not only accelerates *itself*, but also drastically improves the computing efficiency for the *LLM*. We also observe that when reducing the same number of tokens, performing token reduction in the image encoder owns +25.3% throughput boost and -21.2% memory consumption than pruning tokens within the LLM. However, accelerating the image encoder within LVLMs is few explored by recent works. In this paper, we explore how to incorporate the image encoder for acceleration and coordinate it with the LLM to perform thorough acceleration with higher efficiency.

## 3.2 METHOD

As introduced above, in this paper, we try to overcome the limitation of previous methods that only reduce tokens during or within the LLM in LVLMs, which inevitably can't reach higher efficiency. In practice, we observe that aggressively pruning tokens in the image encoder leads to substantial performance degradation, as excessive token removal at early stages discards critical information. This motivates us to develop a more elegant way to intelligently allocates the computational budget across both the image encoder and the LLM. Besides, to avoid beneficial information loss caused by token reduction operations and maintain model performance, we design a new token merging strategy to recycle potentially beneficial information from those redundant tokens. Specifically, as shown in Fig. 3(b), our proposed method, termed as iLLaVA, aims to accelerate the forward pass of both image encoders and LLM to achieve high efficiency. Within the image encoder, it performs token merging between the attention module and the Feed-Forward Network of several encoder blocks to decrease visual tokens. Within the LLM, it performs token merging operations after several specific LLM blocks. Each token merging module will reduce a constant number of image tokens to decrease the required computational burden. By incorporating the image encoder into model acceleration, iLLaVA could notably decrease the image tokens in early network stages with high computing efficiency, and simultaneously lower the computational costs of the subsequent LLM. As reflected in Fig. 3(a), iLLaVA can accurately preserve the informative tokens in both the image encoder and LLM to keep beneficial information. We next introduce each design in detail.

### 3.2.1 TWO-STAGE TOKEN MERGING

We first introduce the token merging process in the image encoder and LLM, respectively. In the image encoder, we insert a token merging module after the attention module of several encoder blocks to reduce image tokens. Given an encoder block consisting of an Multi-Head Attention module

(MHA) and a Feed-Forward Network (FFN), the forward pass with token merging operations can be represented as:

$$x_{out}^v = \text{FFN}(\text{Token\_Merge}(\text{MHA}(x_{in}^v))). \tag{1}$$

Token_Merge denotes the token merging operation, which reduces a constant number of image tokens for each block. After several token merging operations, the image tokens drastically decrease, largely reducing the computational demand of both the subsequent encoder blocks and LLM.

In the LLM, we insert a token merging module between several specific LLM blocks to reduce image tokens. Given one LLM block $K_i$ that is expected to conduct token merging operations, the forward pass can be represented as:

$$x_{out}^t = \text{Token\_Merge}(K_i(x_{in}^t))). \tag{2}$$

After the token merging operations, the number of image tokens gradually decreases, which largely diminishes the computational requirements of parameter-intensive LLM.

We apply the same token merging strategy with different merging tokens for the image encoder and language model, respectively. In the image encoder, we select $B_v$ blocks and merge tokens by $R_v$ per block. In the language model, we perform token merging operations of $B_t$ selected blocks and reduce $R_t$ image tokens per block. Thus, given $N$ input tokens, we finally keep $N - R_v \times B_v - R_t \times B_t$ tokens. The computational budget can be dynamically adjusted by modulating the number of reduced tokens in each stage.

### 3.2.2 TOKEN MERGING STRATEGY

To effectively reduce redundancy while retaining the most informative visual tokens, it is crucial to accurately identify which tokens contribute most to the model's predictions. As we have concluded from the observations in Sec. 3.1, attention scores from the attention module in the image encoder could well reveal the location of informative tokens. We also observe that attention scores in the LLM serve as a good indicator for the importance of each token. We thus use the attention score from the LVLM as our metrics for measuring token importance.

Specifically, the attention scores are calculated as Eq. 3:

$$S_h = \text{Softmax}(\frac{Q_h K_h^T}{\sqrt{D_h}}), \tag{3}$$

where $S_h \in \mathcal{R}^{N \times N}$ is the attention score of each head and we omit the batch dimension here. $D_h$ is the head dimension, and $Q_h$ and $K_h$ represent query and key, respectively.

We take the image encoder as an example to introduce our token merging strategy. Given $N$ input image tokens, the token merging module reduces $R_v$ tokens to decrease required computational costs, only keeping $P_v = N - R_v$ image tokens. The preserved tokens $P_v$ consists of two subsets with different focuses: $P_v^i$ informative tokens, which preserve the most important information of inputs, and $P_v^c$ recycled tokens, which try to recycle beneficial information from the less important tokens. Accordingly, we have $P_v^i + P_v^c = P_v$.

We begin by first selecting the informative tokens $P_v^i$ from inputs that are most important to the model. To verify which tokens are critical, we average $S_h$ across the head dimension over all tokens to obtain an attention map $S_{\text{avg}} \in \mathcal{R}^N$. $S_{\text{avg}}$ indicates the importance of each token to other tokens. Based on $S_{\text{avg}}$, we select top $P_v^i$ tokens with the highest attention scores as $P_v^i$ informative tokens, which are expected to keep the most informative information of visual inputs.

In the above token selection procedure, other tokens are identified as less important ones and will not be preserved in this process. However, these tokens may still contain beneficial information for the model, and directly discarding them may cause beneficial information loss and lead to performance degradation. To avoid performance drop and preserve potentially critical information from these tokens as much as possible, we propose to recycle beneficial information from these tokens by condensing them into some representative tokens. To form robust and information-intensive representations, we assign $P_v^c$ tokens with the highest attention scores based on $S_{\text{avg}}$ (excluding $P_v^i$ informative tokens) as recycled tokens, which are experted to serve as clusters to aggregate potentially beneficial information from other discarded tokens. To obtain the relationships between recycled tokens $P_v^c$ and others, we compute the attention scores between them following Eq. 3 and

Table 1: Results on image benchmarks. RealWorldQA grok 1.5v (2024) is abbreviated as RWQA.

| Method | MMMU (Val) | MMMU Pro Overall | MMBench (EN) | MMBench V1.1 (EN) | MME (sum) | MMStar (Test) | MMVet (Test) | MuirBench | RWQA | Avg |
|---|---|---|---|---|---|---|---|---|---|---|
| Vanilla | 58.6 | 38.3 | 83.5 | 82.6 | 2347 | 63.9 | 67.1 | 59.6 | 68.5 | 65.3 |
| (Upper bound) | 100.0% | 100.0% | 100.0% | 100.0% | 100.0% | 100.0% | 100.0% | 100.0% | 100.0% | 100.0% |
| | | | | Reduce about 66.7% image tokens | | | | | | |
| FasterVLM | 57.9 | 37.7 | 81.2 | 80.8 | 2258 | 61.1 | 63.9 | 57.8 | 66.4 | 63.5 |
| (ICCV25) | 98.8% | 98.4% | 97.2% | 97.8% | 96.2% | 95.6% | 95.2% | 96.8% | 97.0% | 97.0% |
| PyramidDrop | 57.9 | 37.8 | 81.6 | 80.5 | 2267 | 60.9 | 64.3 | 58.7 | 66.5 | 63.7 |
| (CVPR25) | 98.8% | 98.6% | 97.7% | 97.5% | 96.6% | 95.4% | 95.8% | 98.2% | 97.1% | 97.3% |
| SparseVLM | 58.4 | 38.3 | 82.7 | 81.2 | 2299 | 61.9 | 64.6 | 58.1 | 67.2 | 64.2 |
| (ICML25) | 99.6% | 100.0% | 99.0% | 98.4% | 98.0% | 96.8% | 96.4% | 97.4% | 98.2% | 98.2% |
| VisionZip | 58.3 | 38.2 | 82.5 | 82.0 | 2299 | 62.1 | 64.8 | 58.2 | 67.4 | 64.4 |
| (CVPR25) | 100.0% | 99.6% | 98.8% | 99.2% | 98.0% | 97.2% | 96.8% | 97.6% | 98.3% | 98.4% |
| iLLaVA (Ours) | 58.2 | 38.1 | 83.0 | 81.7 | 2328 | 62.8 | 66.0 | 59.1 | 68.2 | **64.8** |
| | 99.2% | 99.5% | 99.4% | 99.1% | 99.2% | 98.3% | 98.4% | 99.2% | 99.6% | **99.2%** |
| | | | | Reduce about 77.8% image tokens | | | | | | |
| PyramidDrop | 55.3 | 36.4 | 78.5 | 77.7 | 2178 | 58.6 | 61.7 | 55.3 | 64.1 | 60.6 |
| (CVPR25) | 94.3% | 95.1% | 94.0% | 93.6% | 92.8% | 91.7% | 91.9% | 92.8% | 93.4% | 93.2% |
| SparseVLM | 56.7 | 36.6 | 80.3 | 78.3 | 2208 | 60.1 | 62.9 | 56.4 | 64.7 | 62.2 |
| (ICML25) | 96.8% | 95.6% | 96.2% | 95.0% | 94.1% | 93.9% | 93.7% | 94.5% | 95.0% | 95.1% |
| FasterVLM | 56.8 | 36.5 | 79.6 | 79.6 | 2210 | 60.5 | 63.7 | 56.3 | 66.0 | 62.9 |
| (ICCV25) | 97.0% | 96.5% | 95.9% | 97.1% | 94.2% | 94.7% | 95.0% | 94.3% | 96.5% | 95.8% |
| VisionZip | 56.7 | 37.5 | 79.5 | 80.7 | 2280 | 61.3 | 63.1 | 57.6 | 64.8 | 63.6 |
| (CVPR25) | 96.6% | 98.0% | 95.8% | 97.9% | 97.3% | 96.0% | 94.9% | 96.6% | 95.2% | 96.4% |
| iLLaVA (Ours) | 57.9 | 37.8 | 82.9 | 79.8 | 2266 | 61.6 | 65.2 | 57.2 | 67.7 | **64.3** |
| | 99.3% | 98.7% | 99.3% | 96.7% | 96.9% | 96.2% | 97.3% | 96.0% | 99.1% | **97.6%** |
| | | | | Reduce about 88.9% image tokens | | | | | | |
| PyramidDrop | 52.3 | 34.0 | 74.4 | 71.9 | 2039 | 55.1 | 57.3 | 53.0 | 59.8 | 57.2 |
| (CVPR25) | 89.2% | 88.7% | 89.1% | 87.3% | 86.9% | 86.2% | 85.4% | 89.0% | 87.4% | 87.5% |
| SparseVLM | 55.6 | 36.2 | 78.2 | 75.8 | 2128 | 58.0 | 60.8 | 54.6 | 64.7 | 60.0 |
| (ICML25) | 94.3% | 94.5% | 93.2% | 92.0% | 91.1% | 90.9% | 90.8% | 91.7% | 94.3% | 92.8% |
| FasterVLM | 55.9 | 36.4 | 78.7 | 77.1 | 2131 | 58.5 | 62.2 | 55.2 | 63.7 | 61.0 |
| (ICCV25) | 94.8% | 95.0% | 93.6% | 94.1% | 91.1% | 91.5% | 92.7% | 92.6% | 93.1% | 93.2% |
| VisionZip | 56.0 | 35.9 | 79.2 | 76.1 | 2159 | 58.8 | 62.0 | 54.8 | 65.4 | 61.4 |
| (CVPR25) | 95.2% | 94.0% | 94.2% | 93.1% | 92.0% | 91.9% | 92.4% | 91.8% | 95.3% | 93.7% |
| iLLaVA (Ours) | 56.9 | 36.6 | 79.7 | 79.7 | 2207 | 60.5 | 63.0 | 56.3 | 65.3 | **62.8** |
| | 96.9% | 97.0% | 95.5% | 96.5% | 93.6% | 93.4% | 93.2% | 94.3% | 94.8% | **95.2%** |

average the results across the head dimension to obtain $S_{\text{sub}} \in \mathcal{R}^{P_v^c \times (R_v + P_v^c)}$. We then aggregate complementary features from other closely correlated tokens into these $P_v^c$ recycled tokens to recycle beneficial information. For each recycled token, we assign $M$ (usually $M = 5$) tokens with the highest similarity as its group members, and merge tokens within the same group by a weighted sum according to their normalized attention scores. Finally, the token recycling operation would keep $P_v^i$ informative tokens and $P_v^c$ cluster tokens, which are concatenated as the outputs of the token merging module.

## 3.3 Efficiency Analysis

**Compatibility with Flash-Attention.** For the vanilla attention, we can obtain pre-calculated attention scores $S_h$ to compute $S_{\text{avg}}$. However, Flash-Attention can't return the full attention matrix $S_h$ during inference. In practice, we pass the parameter $return\_attn\_probs$ to the $flash\_attn\_varlen\_func$ function to enable returning cumsum attention weights $S_{\text{cumsum}} \in \mathcal{R}^N$, which can be transformed to $S_{\text{avg}}$ via a simple transformation, eliminating extra computations.

**Computational Complexity.** For either vanilla attention or Flash-Attention, we can obtain $S_{\text{avg}}$ without incurring extra computations. The additional computations are brought by the calculation of $S_{\text{sub}}$. Theoretically, the only computational costs incurred by this procedure is $O(R_v) \times (B_v) + O(R_t) \times (B_t)$. As $R_v$ and $R_t$ are rather small (tens of tokens) compared to the input tokens (usually thousands of tokens) in practice, the incurred extra computations are few and can be ignored.

## 4 Experimental Results

## 4.1 Model Settings

By default, we use Qwen2.5-VL 7B Bai et al. (2025) as the base model. We allocate 40% of merged tokens to the image encoder, distributing them evenly across Layers 5, 9, and 13. The remaining 60% of merged tokens are assigned to the LLM, evenly distributed across Layers 2, 8, and 14.

Table 2: Results on video benchmarks with Qwen2.5-VL 7B over different token reduction ratios. EgoSchema Mangalam et al. (2023) is abbreviated as EgoS here.

| Method | VideoMME (w/o sub)) | MVBench | EgoS (Test) | MLVU (M-Avg) | Avg | VideoMME (w/o sub)) | MVBench | EgoS (Test) | MLVU (M-Avg) | Avg |
|---|---|---|---|---|---|---|---|---|---|---|
| Vanilla | 71.6 / 65.1 | 69.6 | 65.0 | 70.2 | 68.3 | 71.6 / 65.1 | 69.6 | 65.0 | 70.2 | 68.3 |
| (Upper bound) | 100.0%/100.0% | 100.0% | 100.0% | 100.0% | 100.0% | 100.0%/100.0% | 100.0% | 100.0% | 100.0% | 100.0% |
| | Reduce about 90% video tokens | | | | | Reduce about 95% video tokens | | | | |
| SparseVLM | 67.1 / 60.3 | 66.7 | 61.1 | 66.9 | 64.0 | 66.4 / 59.6 | 65.9 | 60.5 | 65.7 | 63.1 |
| (ICML25) | 93.7%/92.9% | 95.8% | 94.0% | 95.3% | 94.5% | 92.8%/91.6% | 94.7% | 93.0% | 93.5% | 93.2% |
| PyramidDrop | 68.6 / 61.4 | 68.0 | 62.2 | 68.4 | 66.5 | 67.6 / 60.2 | 67.5 | 61.5 | 67.0 | 64.9 |
| (CVPR25) | 95.8%/94.5% | 97.2% | 95.8% | 96.9% | 96.1% | 94.5%/93.2% | 96.8% | 94.6% | 95.1% | 94.8% |
| FasterVLM | 68.6 / 61.6 | 68.3 | 62.3 | 68.2 | 66.6 | 67.8 / 60.9 | 67.6 | 61.8 | 67.1 | 65.3 |
| (ICCV25) | 95.8%/94.7% | 97.6% | 95.9% | 96.6% | 96.2% | 94.8%/93.8% | 96.9% | 95.0% | 95.5% | 95.3% |
| VisionZip | 69.2 / 62.0 | 68.4 | 62.5 | 69.0 | 67.0 | 67.9 / 60.9 | 67.2 | 61.7 | 67.1 | 65.5 |
| (CVPR25) | 96.6%/95.3% | 97.8% | 96.2% | 97.3% | 96.7% | 94.9%/93.6% | 96.4% | 94.9% | 95.4% | 95.1% |
| iLLaVA (Ours) | 69.9 / 63.0 | 68.9 | 63.3 | 69.8 | **68.2** | 68.6 / 62.3 | 67.9 | 62.7 | 69.3 | **66.9** |
| | 97.7%/96.7% | 98.5% | 97.4% | 98.6% | **97.8%** | 96.6%/95.7% | 97.3% | 96.4% | 97.8% | **96.8%** |

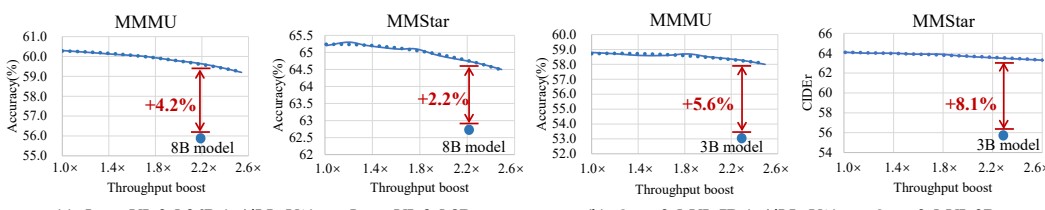

(a) InternVL 2.5 26B (w/ iLLaVA) v.s. InternVL 2.5 8B          (b) Qwen2.5-VL 7B (w/ iLLaVA) v.s. Qwen2.5-VL 3B

Figure 3: (a): deploying iLLaVA upon InternVL2.5 26B and comparing it with InternVL2.5 8B. (b): deploying iLLaVA upon Qwen2.5-VL 7B and comparing it with Qwen2.5-VL 3B.

## 4.2 EFFECTIVENESS ON IMAGE UNDERSTANDING

We compare our method with SparseVLM Zhang et al. (2025b), FasterVLM Zhang et al. (2025a), PyramidDrop Xing et al. (2025) and VisionZip Yang et al. (2025) over a wide range of image understanding benchmarks Liu et al. (2024); Xu et al. (2023a); Yue et al. (2024a;b); Chen et al. (2024b); Yu et al. (2023); Wang et al. (2024a); grok 1.5v (2024) to validate its effectiveness on image understanding. To comprehensively assess performance, we present the results in percentage format for comparative analysis, with the vanilla model's accuracy serving as the 100% upper limit. We use three vision token reduction configurations (66.7%, 77.8% and 88.9%) to evaluate the advantage of our iLLaVA. As shown in Tab. 1, our method demonstrates evident performance advantages over all token reduction ratios compared to other methods. When employing 66.7% token reduction ratios, our method could retain 99.2% performance, owning only a 0.8% decrease compared to the vanilla model. When increasing the token reduction ratios to 77.8% and 88.9%, the advantage of our method is more obvious, outperforming other methods with larger margins upon performance. When reducing 88.9% image tokens, our iLLaVA could still keep 95.2% performance, representing only a 4.8% decrease compared to the vanilla method using 10 times the visual tokens.

## 4.3 EFFECTIVENESS ON VIDEO UNDERSTANDING

We evaluate iLLaVA on four representative video benchmarks including VideoMME Fu et al. (2024), MVBench Li et al. (2024c), Egoschema Mangalam et al. (2023) and MLVU Zhou et al. (2024). We compare our method with SparseVLM Zhang et al. (2025b), FasterVLM Zhang et al. (2025a), PyramidDrop Xing et al. (2025) and VisionZip Yang et al. (2025) upon two heavy token reduction configurations (90% and 95%). We use Qwen2.5-VL 7B as the backbone and set its performance on corresponding video benchmarks as the 100% upper limit. As shown in Tab. 2, our method demonstrates strong superiority compared to existing approaches with either 90% or 95% token reduction ratios. When reducing 90% visual tokens, iLLaVA outperforms the state-of-the-art VisionZip Yang et al. (2025) by 1.2%. When increasing the token reduction ratio to 95%, the performance gap between our method and VisionZip advances to 1.7%, validating its efficacy in handling high reduction ratios.

## 4.4 COMPARISON WITH SMALLER MODELS

An advantage of iLLaVA is that it could enable a larger model to own both better performance and higher efficiency than a smaller model. In Fig. 3, we deploy iLLaVA upon InternVL-2.5 26B Chen et al. (2024c) and Qwen2.5-VL 7B Bai et al. (2025), and compare them with smaller models including InternVL-2.5 8B and Qwen2.5-VL 3B over both accuracy and throughput to validate its effectiveness. As shown in Fig. 3 (a), equipped with iLLaVA, InternVL-2.5 26B could achieve similar or better throughputs than InternVL-2.5 8B, while gaining +4.2% and +2.2% performance advantages on the MMMU and MMStar benchmarks. In Fig. 3 (b), when equipped with iLLaVA, Qwen2.5-VL 7B could achieve higher throughputs than Qwen2.5-VL 3B and obtain +5.6% and +8.1% performance boost on the MMMU and MMStar benchmarks. This reflects that iLLaVA could overcome the computing challenges of large models by enabling larger models to run faster and perform better than smaller models, thus significantly broadening the application scenarios of large models.

## 4.5 FLEXIBILITY OF ILLAVA

Except for Qwen2.5-VL Bai et al. (2025), We deploy iLLaVA upon different LVLMs to verify its flexibility to different model architectures. We adopt LLaVA-Onevision Li et al. (2024a), InternVL-2.5 Chen et al. (2024c) and Minicpm-V 2.6 Yao et al. (2024) as our backbones to conduct a comprehensive evaluation. We compare iLLaVA with previous methods upon four representative image benchmarks including MMMU Yue et al. (2024a), MMBench Liu et al. (2024), MMStar Chen et al. (2024b) and RealWorldQA grok 1.5v (2024) and set the token reduction ratio as 77.8%. Tab. 3 shows iLLaVA consistently demonstrates better averaged accuracy compared to previous state-of-the-art methods when deployed upon different LVLMs, notably validating its flexibility.

Table 3: Flexibility of iLLaVA upon various VLMs. We abbreviate RealWorldQA as RWQA here.

| | MMMU (Val) | MMBench (EN Dev) | MMStar (Test) | RWQA (Test) | Avg |
|---|---|---|---|---|---|
| LLaVA Onevision 7B | 48.8 | 80.8 | 61.7 | 66.3 | 64.4 |
| PyramidDrop (CVPR25) | 46.5 | 76.8 | 59.4 | 62.7 | 61.3 |
| SparseVLM (ICML25) | 46.8 | 77.6 | 59.2 | 63.6 | 61.8 |
| FasterVLM (ICCV25) | 47.6 | 78.8 | **60.4** | 64.6 | 62.8 |
| VisionZip (CVPR25) | 48.0 | 79.2 | 60.2 | 65.1 | 63.2 |
| iLLaVA (Ours) | **48.4** | **79.7** | 60.2 | **65.8** | **63.7** |
| InternVL2.5 8B | 56.0 | 84.6 | 62.8 | 70.1 | 68.4 |
| PyramidDrop (CVPR25) | 52.6 | 80.6 | 59.8 | 65.2 | 64.3 |
| SparseVLM (ICML25) | 53.1 | 81.3 | 59.2 | 65.6 | 64.8 |
| FasterVLM (ICCV25) | 54.2 | 82.5 | 60.2 | 66.8 | 65.9 |
| VisionZip (CVPR25) | 54.8 | **83.2** | 61.3 | 68.7 | 67.0 |
| iLLaVA (Ours) | **55.4** | 83.1 | **61.8** | **69.4** | **67.5** |
| MiniCPM-V2.6 8B | 49.8 | 81.5 | 57.5 | 65.0 | 63.5 |
| SparseVLM (ICML25) | 47.6 | 77.8 | 54.9 | 62.1 | 60.6 |
| PyramidDrop (CVPR25) | 48.0 | 78.6 | 54.7 | 62.7 | 61.0 |
| FasterVLM (ICCV25) | 48.6 | 79.5 | 56.1 | 63.4 | 61.9 |
| VisionZip (CVPR25) | 48.9 | 80.2 | **56.5** | 64.0 | 62.4 |
| iLLaVA (Ours) | **49.3** | **81.0** | 56.3 | **64.3** | **62.8** |

## 4.6 EFFICIENCY ANALYSIS

We validate the efficiency of iLLaVA by comparing it with state-of-the-art methods including SparseVLM Zhang et al. (2025b), FasterVLM Zhang et al. (2025a), PyramidDrop Xing et al. (2025) and VisionZip Yang et al. (2025). These methods are evaluated over various metrics including memory usage, throughput, prefilling time and accuracy across different visual token reduction ratios on the MMMU benchmark. Here, the prefilling time denotes the time required to generate the first output token. As shown in Fig. 4, iLLaVA largely reduces the memory usage by 1.59×, improves the model throughput by 2.12× and reduces the prefilling time by 4.46×. When evaluated upon the same token reduction ratio, iLLaVA not only achieves higher model efficiency than previous methods with lower memory usage and better throughputs, but also obtains better model performance. This indicates that iLLaVA could simultaneously deliver enhanced model accuracy and efficiency than previous methods. This is because iLLaVA leverages the visual redundancy in both the image encoder and LLM to accelerate LVLMs, thus further improving the upper limits. Additionally, we observe that as token reduction ratios gradually increase from 50% to 90%, the advantage of iLLaVA on improving model efficiency and maintaining model accuracy is more obvious. iLLaVA could save more memory usage and achieve higher network throughput than previous methods, while better preserving the model accuracy.

## 4.7 ABLATION STUDY

**Effectiveness of the two-stage design.** In the ablation study, we report the averaged accuracy for iLLaVA and other methods over 9 image understanding benchmarks Liu et al. (2024); Xu et al.

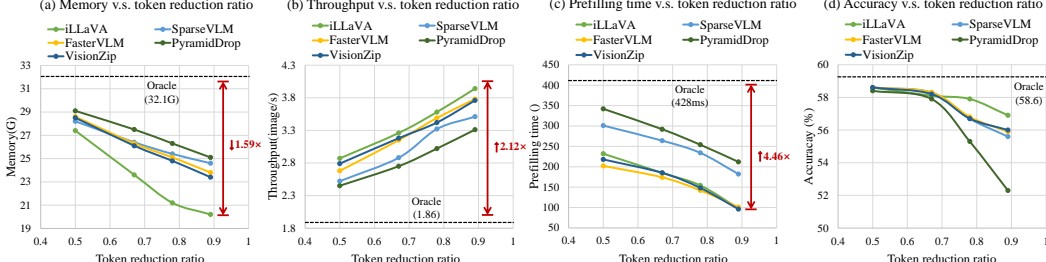

Figure 4: Comparison over memory usage, throughput, prefilling time and accuracy over the MMMU benchmark. Prefilling time denotes the latency of generating the first output token. iLLaVA owns notably lower memory usage and throughputs and retains better accuracy than other methods, while achieving comparative prefilling time.

(2023a); Yue et al. (2024a;b); Chen et al. (2024b); Yu et al. (2023); Wang et al. (2024a); grok 1.5v (2024) by reducing 88.9% visual tokens. We first validate the effectiveness of accelerating LVLMs in the image encoder and LLM stage, respectively. As shown in Tab. 4. We observe that compared to the vanilla model, conducting token reduction in the LLM stage could notably improve the model throughput and decrease the memory usage. When incorporating the image encoder into model acceleration, we could further improve the model throughput by 2.12×, decrease the memory usage to 0.64× while achieving better accuracy. This reflects that conducting token reduction in both the image encoder and LLM is more effective than just reducing tokens in the LLM.

**Effectiveness of our proposed token merging strategy.** We evaluate the effectiveness of our proposed token merging strategy by replacing it with other token reduction strategies and keeping other network configurations unchanged to conduct a fair evaluation. The methods used for comparison include token pruning strategies Xing et al. (2025) and token merging strategies Zhang et al. (2025b); Yang et al. (2025); Shang et al. (2024) from recent representative methods. Except for approaches specially designed for LVLMs, we notice that a previous representative method named ToMe Bolya et al. (2022) has explored reducing tokens in the image encoder, and incorporate it in our experiments. As shown in Tab. 5, compared to other approaches, our proposed strategy achieves better averaged accuracy, which validates the design of recycling beneficial information from discarded tokens. Besides, our strategy is also efficient in technical implementations, which gains comparable or better throughputs compared to state-of-the-art methods.

Table 4: Validation of the two-stage design.

| Encoder | LLM | Acc(%) | Throughput | Memory |
|---|---|---|---|---|
| ✗ | ✗ | 65.3 | 1.86 | 32.1G |
| ✗ | ✓ | 62.1 | 3.46 | 23.1G |
| ✓ | ✓ | **62.8** | **3.94 (2.12×)** | **20.6G (0.64×)** |

Table 5: Comparing different token reduction ways.

| Style | Methods | Acc(%) | Throughput |
|---|---|---|---|
| Pruning | PyramidDrop | 61.0 | 4.63 |
| Token merging | Bipartite matching | 56.4 | 4.50 |
| | LLaVA-PruMerge | 61.1 | 4.52 |
| | SparseVLM | 61.7 | 4.60 |
| | VisionZip | 62.1 | 4.61 |
| | iLLaVA | **62.8** | 4.62 |

Table 6: Comparison with more training-free token reduction methods.

| Style | Methods | Reduction ratio | |
|---|---|---|---|
| | | 77.8% | 88.9% |
| Token pruning | FastV | 58.2 | 52.4 |
| | PyramidDrop | 62.2 | 57.2 |
| | AdaFV | 62.4 | 59.8 |
| | FasterVLM | 62.9 | 61.0 |
| | DivPrune | 63.1 | 60.8 |
| | FEATHER | 63.4 | 61.0 |
| Token merging | LLaVA-PruMerge | 61.8 | 59.6 |
| | SparseVLM | 62.2 | 60.0 |
| | VisionZip | 63.6 | 61.4 |
| | AIM | 63.8 | 61.2 |
| | iLLaVA | **64.3** | **62.8** |

**Comparison with more token reduction methods.** We compare our iLLaVA with more token reduction methods Chen et al. (2024a); Xing et al. (2025); Han et al. (2025); Zhang et al. (2025a); Alvar et al. (2025); Endo et al. (2024); Shang et al. (2024); Zhang et al. (2025b); Yang et al. (2025); Zhong et al. (2024) designed for LVLMs to confirm its superiority. We perform the experiments under two token reduction ratios including 77.8% and 88.9% and show the results in Tab. 6. We observe that while some token pruning methods own promising performance, token merging methods generally achieve better results compared to token pruning methods. Compared to these state-of-the-art approaches, iLLaVA achieves the best performance across both token reduction ratios, validating its superiority among these strong competitors.

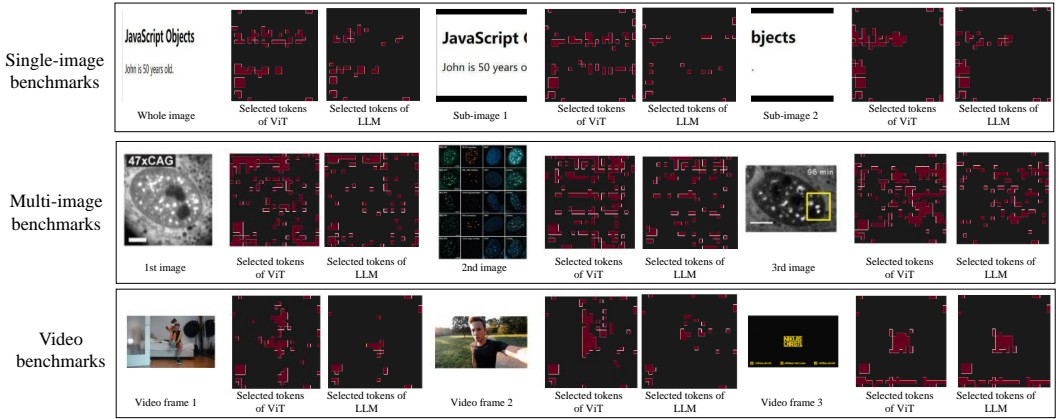

Figure 5: Visualizations for selected tokens of iLLaVA over different benchmarks. We select several representative images for each benchmark, and display the raw image, selected tokens from the image encoder and selected tokens from the LLM in each row in order.

## 4.8 VISUALIZATIONS

To better understand the token merging process in iLLaVA, we visualize the selected image tokens for single-image benchmarks, multi-image benchmarks and video benchmarks in Fig. 4.7, respectively. On each benchmark, we select three images and visualize the raw image, selected tokens from the image encoder and selected tokens from the LLM in order. We observe that both the image encoder and LLM can well attend to the important information from raw images, like the words in the first row and the person in the last row. We also notice that the LLM tends to allocate more tokens to images closer to the output texts, and pay less attention to images which are input earlier into the LLM. This may affect the behavior of LVLMs in memorying long contexts.

## 4.9 COMPUTATION DISTRIBUTION IN IMAGE ENCODER AND LLM

To accelerate LVLMs by reducing computations in both the image encoder and LLM, a key challenge is allocating the computation reduction between the two components. Fig. 6 shows how performance and throughput scale with the proportion of computation reduced in the image encoder. When we reduce 0-40% of computations in the image encoder (and 100–60% in the LLM), performance only drops slightly while throughput improves significantly ($1.8\times$ to $2.1\times$). However, further increasing this proportion to 40-90% leads to a rapid performance decline, even as throughput grows more sharply. This indicates that moderate reduction of image tokens (0-40%) has little impact on performance, but excessive reduction degrades it substantially by stripping the fine-grained spatial information needed by the model.

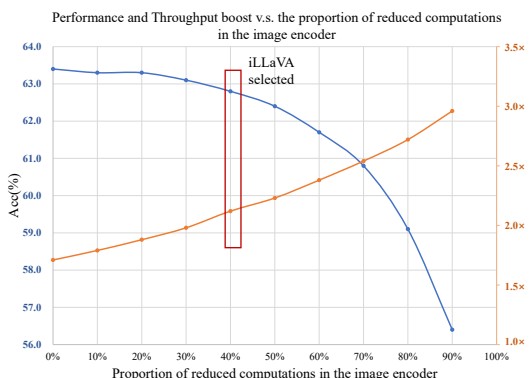

Figure 6: Performance and Throughput boost v.s. the proportion of reduced computations in the image encoder.

We also find reducing image tokens in the encoder boosts throughput more effectively than reducing tokens in the LLM, since the encoder contributes most of the input tokens to the LLM. Based on these observations, we select a 40% reduction in the image encoder and a 60% reduction in the LLM to balance performance and efficiency.

## 5 CONCLUSION

In this paper, we review and validate that there exists considerable feature redundancy in the image encoder of LVLMs. We propose to utilize and combine it with the visual redundancy in other LVLM components (e.g., LLM) to accelerate models for further efficiency. Moreover, to avoid performance loss caused by token reduction, we propose a new token merging strategy to recycle the potentially beneficial information from discarded tokens. Results on 10+ image and video understanding benchmarks across various token reduction ratios validate the effectiveness of our method.

## ACKNOWLEDGEMENTS

Our work was supported by the following projects: National Natural Science Foundation of China (Grant No. U2574216); Emerging Frontiers Cultivation Program of Tianjin University Interdisciplinary Center; National Natural Science Foundation of China (No. 62276182 and 62572349).

## ETHICS STATEMENT

This work adheres to the ICLR Code of Ethics and has been conducted with careful consideration of ethical implications. The research does not involve human subjects or personally identifiable data, and no new datasets requiring special release protocols were created or used. All methodologies and applications presented are intended for constructive, non-harmful purposes, and we have taken steps to identify and mitigate potential biases in model design and training data. No conflicts of interest exist, and the research received no industry sponsorship that could influence outcomes. Privacy, security, and legal compliance were prioritized throughout the experimental process. We uphold principles of research integrity through transparent documentation, reproducible experiments, and responsible dissemination of results.

## REPRODUCIBILITY STATEMENT

To ensure the reproducibility of our results, we have taken comprehensive steps to document and share all necessary components of our work. The model architecture, training procedures, and hyperparameter settings are fully described in the implementation details of the main paper. All datasets used in our experiments are publicly available. The code of our work will be made publicly available if this paper receives acceptance by ICLR2026.

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

## A  APPENDIX

### A.1  LIMITATIONS

While we validate that iLLaVA performs well across a wide range of benchmarks, we find that upon benchmarks like DocVQA, InfoVQA and ChartQA which require a large quantity of image tokens to perform detailed document understanding, conducting token merging may lead to more evident performance drop. Moreover, image images may contain small objects that are difficult for the models to capture and build fine-grained features, which are easily missed during the token reduction process. In these scenarios, the model could demonstrate inferior performance compared to the vanilla model as they lack the critical information contained in the small objects for understanding.

### A.2  LLM USAGE

In this work, we employed large language models (LLMs) solely for grammatical refinement and language polishing. No substantive content, arguments, or technical formulations were generated or altered by the LLM; its use was strictly limited to correcting syntax, improving sentence fluency, and ensuring grammatical accuracy in the final manuscript. All intellectual contributions, including conceptual development, analysis, and writing, remain entirely the authors' own.

