# OpenReview forum: "iLLaVA: An Image is Worth Fewer Than 1/3 Input Tokens in Large Multimodal Models"
_ICLR.cc/2026/Conference — ICLR 2026 Poster_

### Official Review · Reviewer_MrAM · 2025-10-29

**Soundness:** 2
**Presentation:** 3
**Contribution:** 2
**Rating:** 6
**Confidence:** 2

**Summary:**

This work proposes iLLaVA, a method that accelerates VLMs with token reduction. Different from most prioir work that focues on reducing in only LLMs, this work considers both the image encocer and the LM. Moreover, a merging strategy is further introduced to include potentially useful information. The proposed approach is evaluated on a variety of tasks and models, and the results illustrate the its effectiveness: it can obtain over 95% of the performance over various token reduction ratios and perform better than other methods.

**Strengths:**

- The direction of accelerating VLMs is important for the real-world deployment.
- The idea of joint optimization of both image encoder and LLM is intuitive.
- The results illustrate the effectiveness of the proposed approach.

**Weaknesses:**

- Using the attention score as an indicator of token importance is not quite a novel idea; in addition to applying the reduction to both vision encoder and LM, the methodological differences to previous work should be further illustrated.
- There seems to be a lack of detailed analyses on why the proposed method works well? It will be better to provide more detailed analyses to explain the better performance compared to other methods.
- It seems not quite straight-forward for me to understand the token merging process. There should be more detailed illustration for this process.

**Questions:**

(* Please refer to the weakness part.)

---

> ### Author Response · Authors · 2025-11-22
> **Response for Reviewer MrAM**
>
> 1. The methodological differences to previous works.
>
> Thanks for your question. As input tokens contain considerable feature redundancy for VLMs, it’s natural to reduce redundant tokens in the forward pass of VLMs to largely improve model efficiency. Previous methods usually try to directly prune redundant tokens by directly discarding them or merge them with preserved token with simple aggregations to achieve token reduction. This easily leads to beneficial information loss and bring considerable performance drop. Thus, except from trying to allocate token reduction budgets to both the ViT/LLM stage to achieve a good accuracy-efficiency trade-off, in this paper we also propose a novel token merging algorithm to recycle beneficial information from unselected tokens to minimize performance drop. Specifically, we first divide the preserved tokens into two subsets. The first set of preserved tokens (informative tokens) is selected based on the averaged attention scores, which only keep input visual tokens with the highest importance. The second set of preserved tokens (recycled tokens) is the recycled part of the unselected tokens in the above selection process. The unselected tokens are not discarded. Instead, they are clustered based on similarity, and each cluster is condensed into a representative recycled token by aggregating features within each cluster group. Finally, the informative tokens and the recycled cluster tokens are concatenated to form a compact yet information-preserving token set for next layer processing. Our algorithm is also made to be compatible with flash attention which overcomes the drawback of many previous works that can’t be applied to flash attention.
>
> 2. Lack of detailed analyses on why the proposed method works well.
>
> Thanks, the performance superiority of our method comes from two perspectives. First, allocating token reduction budgets to both the ViT and LLM leads to better performance than only reducing tokens in a single stage (e.g., LLM) adopted by most previous methods. As shown in the following results, when keeping the number of final preserved tokens unchanged and only varying the location of token reduction operations, performing token reduction in both the ViT and LLM stage would lead to a better accuracy-efficiency trade-off than just performing token reduction in the LLM stage.
>
> | Encoder | LLM | Acc(%) | Throughput | Memory |
> |---------|-----|--------|------------|---------|
> | ✘       | ✘   | 65.3   | 1.86       | 32.1G   |
> | ✔       | ✘   | 61.3  | **4.48**    | **17.9 G**   |
> | ✘       | ✔   | 62.1   | 3.46       | 23.1G   |
> | ✔       | ✔   | **62.8** | 3.94 (2.12×) | 20.6G (0.64×) |
>
> Second, the performance advantage of our method also comes from the superiority of our proposed novel token merging strategy. As shown in the following results, when keeping other components unchanged and only varying the token reduction algorithm, our method could achieve superior performance than previous token pruning/merging methods, demonstrating that our method is able to preserve more beneficial information and better maintain the performance.
> | Style | Methods | Reduction ratio (77.8%) | Reduction ratio (88.9%) |
> | :--- | :--- | :--- | :--- |
> | Token pruning | FastV | 58.2 | 52.4 |
> | | PyramidDrop | 62.2 | 57.2 |
> | | AdaFV | 62.4 | 59.8 |
> | | FasterVLM | 62.9 | 61.0 |
> | | DivPrune | 63.1 | 60.8 |
> | | FEATHER | 63.4 | 61.0 |
> | Token merging | LLaVA-PruMerge | 61.8 | 59.6 |
> | | SparseVLM | 62.2 | 60.0 |
> | | VisionZip | 63.6 | 61.4 |
> | | AIM | 63.8 | 61.2 |
> | | iLLaVA | **64.3** | **62.8** |

---

> ### Author Response · Authors · 2025-11-22
> **Response for Reviewer MrAM**
>
> 3. More detailed illustration for the token merging process.
>
> We present more details for our token merging process by decomposing it into multiple steps for better illustration. The finally preserved tokens are consisting of two subsets, informative tokens and recycled tokens.
>
> Step 1: Importance Verification
>
>  Calculate the global attention map $S_{\text{avg}} \in \mathbb{R}^N$ by averaging the input attention scores $S_h$ across the head dimension for all tokens. $$S_{\text{avg}} = \text{Mean}(S_h)$$
> (Note: $S_{\text{avg}}$ represents the importance of each token relative to others.)
>
> Step 2: Select Informative Tokens ($P_v^i$)
>
> (1) Sort tokens based on scores in $S_{\text{avg}}$ in descending order.
>
> (2) Select the top $P_v^i$ tokens with the highest scores.
>
> (3) Store these as Informative Tokens ($P_v^i$).
> $$T_{\text{info}} \leftarrow \text{TopK}(T, S_{\text{avg}}, k=P_v^i)$$
>
> (4) Identify the remaining tokens as the "Unimportant Pool" ($T_{\text{remain}}$).
> (Size of $T_{\text{remain}}$ is $R_v + P_v^c$).
>
> Step 3: Select Recycled Token Candidates ($P_v^c$)
>
> (1) From the "Unimportant Pool" ($T_{\text{remain}}$), identify the top $P_v^c$ tokens with the highest remaining $S_{\text{avg}}$ scores.
>
> (2) Designate these as the Recycled Tokens (Cluster Centers).
> $$T_{\text{recycle}} \leftarrow \text{TopK}(T_{\text{remain}}, S_{\text{avg}}, k=P_v^c)$$
>
> Step 4: Token Recycling (Clustering & Merging)
>
> (1) Compute Relationships: Calculate attention scores between the Recycled Tokens ($T_{\text{recycle}}$) and the full Unimportant Pool ($T_{\text{remain}}$).
>
> (2) Average these scores across the head dimension to obtain the sub-attention map:
> $$S_{\text{sub}} \in \mathbb{R}^{P_v^c \times (R_v + P_v^c)}$$
>
> (3) Grouping: For each Recycled token in $T_{\text{recycle}}$:
>
> ① Identify the $M$ most similar tokens from the Unimportant Pool based on $S_{\text{sub}}$.
>
> ② Form a group containing the Recycled token and its $M$ neighbors.
>
> (4) Merging: Update each Recycled token by calculating a weighted sum of the tokens within its group.
>
> ① Weights are determined by normalized attention scores.
>
> ② This aggregates complementary features from discarded tokens into the Recycled tokens.
>
> Step 5: Final Concatenation
>
>  Concatenate the preserved Informative Tokens and the updated Recycled Tokens to form the output.
> $$T_{\text{out}} = \text{Concat}(T_{\text{info}}, T_{\text{recycle}})$$

---

### Official Review · Reviewer_jbVu · 2025-10-30

**Soundness:** 3
**Presentation:** 3
**Contribution:** 3
**Rating:** 6
**Confidence:** 4

**Summary:**

The paper proposes iLLaVA to address a key limitation of existing LVLM acceleration methods: most only reduce image tokens in the LLM stage, overlooking the image encoder—a critical bottleneck accounting for 99% of inference time (45% in video tasks)—and failing to achieve end-to-end acceleration. To tackle this, iLLaVA enables comprehensive acceleration through two core innovations: two-stage token merging (inserting modules at critical positions in both the image encoder and LLM to jointly optimize the two computationally intensive components) and a novel token merging strategy that recycles useful information from discarded tokens, compatible with Flash-Attention with negligible extra computation. Experiments demonstrate iLLaVA retains over 95% performance while reducing 66.7%-88.9% of image tokens, outperforms SOTA methods, supports multiple LVLM architectures, and delivers significant improvements in memory usage, inference speed, and throughput.

**Strengths:**

- Simple training-free approach. The paper adopts a training-free token merging method for ViT and LLM. Compared with merely merging or pruning tokens on MLLMs, the proposed method achieves better performance, lower memory overhead, and higher throughput.
- Extensive experiments. The paper conducts extensive experiments on two benchmark suites to verify the effectiveness of the proposed method across 9 image tasks and 8 video tasks. Under fair comparison settings, the proposed method achieves better performance than four existing methods.
- High generalization. It proves adaptable to four MLLMs, demonstrating its strong generalization capability.

**Weaknesses:**

- Lack of novelty. The paper adopts the existing token merging method and extends its application from LLM to ViT.
- Simplified visualization in Figure 2. Figure 1 presents the attention scores in ViT for a single object against a simple, plain background. This straightforward case fails to convince readers. What would the attention map look like when the image scene is complex?
- Limited experimental benchmarks. The paper only conducts experiments on general QA benchmarks. How does it perform on benchmarks focusing on capturing key objects in dense information? For instance, text-centric benchmarks (TextVQA, ChartVQA, DocVQA, InfoVQA, OCRBench) and high-resolution image benchmarks (V*, etc.)?
- Typos: Figure 3(b) contains a typo: "QWen2.5-VL 7B" should be formatted as "Qwen2.5-VL".

**Questions:**

- Flexibility of insertion position. The token merging module can be inserted into any layer of ViT and LLM. Do different VLMs require distinct insertion positions of the token merging module to achieve optimal performance? Additionally, are there differences in the optimal insertion positions across various VLMs?
- In Table 4, reducing tokens in both the encoder and LLM achieves better accuracy than only reducing tokens in LLM. Why does reducing more tokens in ViT further improve performance? Moreover, the paper lacks an ablation study focusing on applying the token merging module solely in ViT.

---

> ### Author Response · Authors · 2025-11-22
> **Response for Reviewer jbVu**
>
> 1. Discussion of novelty.
> Thanks for your question. Beyond exploring reducing tokens in both the ViT/LLM stages to reduce information redundancy, we also introduce a new token merging strategy. Previous methods usually try to directly prune redundant tokens by simply discarding them or merge them with preserved token with simple aggregations to achieve token reduction. Instead, we design a novel token merging strategy which recycles beneficial information from discarded tokens to minimize information loss brought by token reduction. It divides preserved tokens into two subsets, with the first subset containing the most important tokens of the highest attention scores and the second subset consisting of tokens with recycled information from redundant tokens. The recycling algorithm first identifies token clusters by grouping tokens with similar semantics into the same group, and then aggregates distinctive features within each group to assign representative features for cluster tokens. Results in Tab.6 in our paper shows that our proposed token merging strategy demonstrates clear superiority compared to existing token pruning/merging strategies. We paste the results here for your convenience.
> | Style | Methods | Reduction ratio (77.8%) | Reduction ratio (88.9%) |
> | :--- | :--- | :--- | :--- |
> | Token pruning | FastV | 58.2 | 52.4 |
> | | PyramidDrop | 62.2 | 57.2 |
> | | AdaFV | 62.4 | 59.8 |
> | | FasterVLM | 62.9 | 61.0 |
> | | DivPrune | 63.1 | 60.8 |
> | | FEATHER | 63.4 | 61.0 |
> | Token merging | LLaVA-PruMerge | 61.8 | 59.6 |
> | | SparseVLM | 62.2 | 60.0 |
> | | VisionZip | 63.6 | 61.4 |
> | | AIM | 63.8 | 61.2 |
> | | iLLaVA | **64.3** | **62.8** |
>
> 2. Simplified visualization in Figure 1.
>
> Many thanks for your key insight! We have replaced Fig.1 with a more complex figure and plot the attention scores of different ViT layers to explore whether ViT could inherently pay attention to important regions of complex input images. You can refer to the Fig.1 of updated pdf file for visualization. The updated input image contains two boys playing football in the foreground and a boy walking in the background in the distance. We observe that in shallow layers, ViT could clearly attend to the two boys in the foreground, and could even pay attention to the boy far away (even it looks smaller in the distance) and the football. This phenomenon quickly diminishes as layers go deep, which demonstrates only attention scores in shallow ViT layers serve as good indicators for important tokens.
>
> 3. Performance on benchmarks requiring dense information.
>
> We compare our method with previous methods upon several text-centric benchmarks including TextVQA, ChartVQA, DocVQA and InfoVQA to verify its effectiveness on understanding images with dense information. We first observe that compared to the oracle model (Qwen2.5-VL), most efficient methods lead to a considerable performance drop, which can be attributed to the inherent drawback of token-reduction-based efficient VLMs which may lose details in the token reduction process. Compared to previous methods, our method can largely relieve the performance drop caused by token reduction and outperform them by a large margin. This demonstrates the advantage of our novel token merging strategy and two-stage token merging design in minimizing the information loss during inference to better preserve performance.
>
> | Methods | TextVQA | ChartVQA | DocVQA | InfoVQA |
> | --- | --- | --- | --- | ---|
> | Qwen2.5-VL 7B (upper bound) | 84.9 | 87.3 | 95.7 | 82.6 |
> | PyramidDrop (CVPR25) | 81.1| 83.7 | 90.2 | 78.8 |
> | SparseVLM (ICML25) | 80.2| 81.4 | 88.6 | 78.1 |
> | FasterVLM (ICCV25) | 80.8| 82.5 | 89.4 | 77.5 |
> | VisionZip (CVPR25) | 81.4 | 83.2 | 90.7 | **79.3** |
> | iLLaVA (ours) | **81.9**| **84.6** | **91.6** | 79.2 |
>
> 4. Typos of words.
>
> Thanks for your advice. We have corrected all expressions in our paper into “Qwen2.5-VL” to keep consistency.

---

> ### Author Response · Authors · 2025-11-22
> **Response for Reviewer jbVu**
>
> 5. Flexibility of insertion position.
>
> Thanks for your question. As shown in the appendix, we first perform a manual search for the optimal insertion places in the ViT/LLM of our token merging modules based on Qwen2.5-VL. We then find that the obtained configurations can be easily transferred across different backbones like Qwen2.5-VL, InternVL2.5 and MiniCPM-V2.6 **without further modifications**. In Tab.3 of our paper, we report the performance of our method across various VLMs without modifying the insertion places, which achieve promising generalizability over different architectures. We list the results in the following for your reference. We observe that iLLaVA could show superior generalizability across different benchmarks with default configurations and notably outperform previous methods. This could be attributed to that mostly current VLMs adopt a similar architecture consisting of a vision encoder, a MLP and a LLM, which makes our model easily transfer across different VLMs. Moreover, considering most recent VLMs usually adopt the same vision encoder (e.g., SigLip) and the same VLM series (e.g., Qwen series), it’s easy for our model to adapt to different VLMs without further modifications.
>
> | Model                         | MMMU | MMBench | MMStar | RealWorldQA | Avg  |
> |------------------------------|------------|-------------------|----------------|--------------|------|
> | Qwen2.5-VL           |  58.6      | 83.5              | 63.9           | 68.5         | 68.6 |
> | PyramidDrop (CVPR25)         | 57.9       | 81.6             | 60.9           | 62.7         | 65.8 |
> | SparseVLM (ICML25)           | 58.4       | 82.7              | 61.9          | 67.2       | 67.6 |
> | FasterVLM (ICCV25)           | 57.9       | 81.2              | 61.1      | 66.4         | 66.7 |
> | VisionZip (CVPR25)           | **58.3**       | 82.5             | 62.1          | 67.4         | 67.6 |
> | **iLLaVA (Ours)**            | 58.2   | **83.0**          | **62.8**           | **68.2**     | **68.1** |
> ---
> | Model                         | MMMU| MMBench | MMStar | RealWorldQA | Avg  |
> |------------------------------|------------|-------------------|----------------|--------------|------|
> | InternVL2.5 8B               | 56.0       | 84.6              | 62.8           | 70.1         | 68.4 |
> | PyramidDrop (CVPR25)         | 52.6       | 80.6              | 59.8           | 65.2         | 64.3 |
> | SparseVLM (ICML25)           | 53.1       | 81.3              | 59.2           | 65.6         | 64.8 |
> | FasterVLM (ICCV25)           | 54.2       | 82.5              | 60.2           | 66.8         | 65.9 |
> | VisionZip (CVPR25)           | 54.8       | **83.2**          | 61.3           | 68.7         | 67.0 |
> | **iLLaVA (Ours)**            | **55.4**   | 83.1              | **61.8**       | **69.4**     | **67.5** |
> ---
> | Model                         | MMMU | MMBench | MMStar | RealWorldQA | Avg  |
> |------------------------------|------------|-------------------|----------------|--------------|------|
> | MiniCPM-V2.6 8B              | 49.8       | 81.5              | 57.5           | 65.0         | 63.5 |
> | SparseVLM (ICML25)           | 47.6       | 77.8              | 54.9           | 62.1         | 60.0 |
> | PyramidDrop (CVPR25)         | 48.0       | 78.6              | 54.7           | 62.7         | 61.0 |
> | FasterVLM (ICCV25)           | 48.6       | 79.5              | 56.1           | 63.1         | 61.4 |
> | VisionZip (CVPR25)           | 48.9       | 80.2              | **56.5**       | 64.0         | 62.4 |
> | **iLLaVA (Ours)**            | **49.3**   | **81.0**          | 56.3           | **64.3**     | **62.8** |
>
> 6. Discussion for results in Tab.4.
>
> Sorry for misclarifications. In Tab.4, we perform experiments by keeping the number of final preserved tokens unchanged, and only change of location (ViT/LLM) to perform token merging. We further provide an ablation to showcase the effects of applying token merging solely in ViT. As shown in the following results, either only decreasing visual tokens in the ViT stage or the LLM stage would lead to worse results. Reducing visual tokens in the ViT stage only would hurt the performance more due to losing beneficial information in early stages. By combining token reduction operations in two stages, we could achieve a good accuracy-efficiency trade-off with promising performance and throughput.
>
> | Encoder | LLM | Acc(%) | Throughput | Memory |
> |---------|-----|--------|------------|---------|
> | ✘       | ✘   | 65.3   | 1.86       | 32.1G   |
> | ✔       | ✘   | 61.3  | **4.48**    | **17.9 G**   |
> | ✘       | ✔   | 62.1   | 3.46       | 23.1G   |
> | ✔       | ✔   | **62.8** | 3.94 (2.12×) | 20.6G (0.64×) |

---

### Official Review · Reviewer_AUQc · 2025-11-01

**Soundness:** 2
**Presentation:** 3
**Contribution:** 2
**Rating:** 4
**Confidence:** 4

**Summary:**

This paper proposed a two-stage method for reducing visual tokens in VLMs. In particular, it tries to reduce tokens both at visual encoder stage and LLM decoding stage. In each stage, a token merging module is inserted in between the self-attention layer and FFN every N blocks. Within the token merging module, tokens that received highest attention scores are deemed to be important and kept, and the rest of tokens are merged into clusters called recycled tokens. Both important and recycled tokens are sent to the next layer for processing. Each token merging module reduces the overall number of tokens by a fixed amount, and the overall model can reduce the visual tokens to the target budget. Experiments on a set of image and video understanding benchmarks with several popular VLMs show that the proposed method outperforms several baselines, both in terms of the performance retention and efficiency.

**Strengths:**

Unlike many previous works in the area that focus on single image task only. This paper presented comprehensive experiments on multi-image and video benchmarks and shows stronger performance compared to several baselines.

The experiments with four different VLMs show the effectiveness of the iLLaVA.

The results on memory usage, prefilling time and thoughput provide interesting insights on the impact of visual token pruning from different angles, which is often neglected in previous works.

**Weaknesses:**

My main concern is on the novelty of this work. The paper claims the two-stage pruning method and token merging as novel contributions. However this idea has been done by previous works. For example, VScan (Zhang et al. 2025) adopt very similar idea to prune tokens at both visual encoder stage and llm decoder stage. Furthermore, VScan also proposed to merge pruned tokens instead of discarding them. The only difference might be the specific layer index where the token pruning/merging happens.

Zhang, Ce, Kaixin Ma, Tianqing Fang, Wenhao Yu, Hongming Zhang, Zhisong Zhang, Yaqi Xie, Katia P. Sycara, Haitao Mi and Dong Yu. “VScan: Rethinking Visual Token Reduction for Efficient Large Vision-Language Models.” ArXiv abs/2505.22654 (2025): n. pag.

**Questions:**

Can authors discuss how the proposed method differs from existing two-stage visual token pruning methods, and showcase the advantage of iLLaVA in experiments?

---

> ### Author Response · Authors · 2025-11-22
> **Response for Reviewer AUQc**
>
> 1. Discussion of existing two-stage pruning methods (e.g., VScan).
>
> Many thanks for your question! We notice that VScan also adopts a two-stage pruning/merging design. We want to highlight the key differences between our iLLaVA and VScan as follows:
>
> (1) Methodology design. Though VScan and ours both employ a two-stage token reduction design, the methodology and intuition are different. In the **visual encoding** stage, VScan try to keep features of coarse and fine granularities, and thus perform token reduction from a global and local perspective, respectively. It reduces visual tokens at the penultimate layer (global scan) and 6th layer (local scan) of the visual encoder. In the LLM, VScan just prunes a part of tokens at a single middle layer. Instead, we try to explore a comprehensive way to allocate computing budgets to different stages of VLMs to minimize performance drop, which can serve as the basis for following methods. Thus, we adopt a progressive design which gradually merges tokens at different intermediate layers by only merging a small fraction of tokens per step. In this way, our methodology is more compatible with the inherent coarse-to-fine feature distribution in VLMs, and bring much smaller performance drop compared to VScan by adopting frequent but minimal token merging operations.
>
> (2) Token pruning/merging strategy. VScan and ours adopt different ways to reduce visual tokens. In the visual encoder, VScan employ token merging to reduce tokens. It simply merges unselected tokens with preserved tokens of the highest similarity by average pooling, which easily hurt the generated representations. In the LLM, it prunes a part of tokens at one middle LLM layer by discarding unselected tokens. This may lead to beneficial information loss in the forward pass. Instead, we design a novel token merging strategy which recycles beneficial information from discarded tokens to minimize information loss brought by token reduction. It divides preserved tokens into two subsets. The first subset contains the most important tokens of the highest attention scores to keep information intensity. The second subset consists of tokens with recycled information from redundant tokens. The recycling algorithm first identifies token clusters by grouping tokens with similar semantics into the same group within the unselected tokens. It then aggregates distinctive features within each group by a weighted sum to assign representative features for cluster tokens. These two subsets will be concatenated as the outputs of the token merging process. As shown in the following experiments, our strategy could better preserve performance compared to VScan during inference.
>
> (3) Compatibility with flash attention. VScan may not be compatible with flash attention. In the vision encoder, VScan relies on the attention score of [CLS] token to identify important tokens. However, as commonly-used modern visual encoders (e.g., SigLip) don’t have a [CLS] token, it has to refer to the averaged outputs of full attention matrix $R^{N \times N}$ to distinguish important tokens. As flash attention doesn’t support return the full attention matrix, VScan may not be compatible with flash attention in the forward procedure. Instead, our token merging process only relies on the sliced attention score $S_{avg}\in R^N$ to distinguish token clusters and perform information recycling, which can be obtained via passing $return\ attn\ probs=True$ to the function $flash\ attn\ varlen\ func()$ in the forward pass of flash attention. Thus, our algorithm is inherently compatible with flash attention.
>
> We conduct a series of experiments to verify the superiority of various components in our iLLaVA compared to VScan based on Qwen2.5-VL 7B. We compare iLLaVA with VScan across MMMU, MMBench, MMStar and RealWorldQA benchmarks. As shown in the following results, we first observe iLLaVA could notably outperform VScan by around 1.0% on these four benchmarks, demonstrating its superiority brought by methodology design. We then verify the effectiveness of different components in iLLaVA, by changing the token reduction design in the visual encoder (denoted as visual design degradation), the token reduction design in the LLM (denoted as LLM design degradation) and the token merging strategy (denoted as token merging strategy degradation) to the configurations of VScan. We notice that changing each component from the design of iLLaVA to VScan would degrade the performance notably, which verifies the effectiveness of each component design in iLLaVA.
>
> | Methods | MMMU | MMBench| MMStar | RealWorldQA|
> | --- | --- | --- | --- | --- |
> | iLLaVA | **56.9** | **79.7** | **60.5** | **65.3** |
> | iLLaVA (Visual design degradation) | 56.1| 79.0 | 59.8 | 64.6 |
> | iLLaVA (LLM design degradation) | 56.4| 79.2 |60.0 | 64.9 |
> | iLLaVA (Token merging strategy degradation) | 56.3 | 79.1 | 59.8| 64.8 |
> | VScan | 55.7| 78.8 | 59.5 | 64.3 |

---

### Official Review · Reviewer_hqY7 · 2025-11-02

**Soundness:** 3
**Presentation:** 3
**Contribution:** 3
**Rating:** 4
**Confidence:** 3

**Summary:**

The paper proposes iLLaVA, achieves a balance between efficiency and accuracy by jointly merging tokens at both the image encoder and LLM ends and introducing "recycled tokens" to aggregate the information of discarded tokens. Despite a 88.9% reduction in visual tokens, it still maintains 95.2% performance, enabling the 26B model to comprehensively outperform the 8B model in terms of speed and efficiency. It offers 2× throughput, 4× first-token latency and 1.59× memory savings, and is suitable for a wide range of LVLMS

**Strengths:**

- The method extends token reduction from LLM to image encoders, achieving dual acceleration and significantly reducing overall computational and memory overhead.

- It aggregates discarded information with "recycled tokens", maintaining an accuracy of over 95% even at extremely high compression rates, balancing speed and performance.

**Weaknesses:**

- Can this method be adapted to flash attention? And, can it be adapted to vllm and sglang inference frameworks? It seems the adaption on mainstream frameworks somehow has difficulties.

- The performance drops significantly for tasks that require fine spatial information, such as DocVQA and ChartQA. Small targets or dense text are prone to losing key details due to token merging.

- The reduction ratio of tokens and the insertion layer positions need to be manually optimized, lacking an adaptive mechanism. When changing models or tasks, it may be necessary to re-search for the optimal configuration, resulting in high migration costs.

**Questions:**

N/A

---

> ### Author Response · Authors · 2025-11-22
> **Response for Reviewer hqY7**
>
> 1. Generalizability with flash attention and vllm inference frameworks.
>
> Thanks for your key questions. As we have introduced in Sec.3.3, our method is compatible with flash attention operations. Flash attention has the inherent constraint which can’t return full attention score matrix during the inference process, which is not compatible with many efficient methods that require the full attention matrix to identify important tokens. Practically, our method only needs the averaged attention score $S_{avg}\in R^N$ to complete the token merging process to identify less important tokens and recycle beneficial information, which avoids this limitation. We could pass the parameter $return\ attn\ probs=True$ to the $flash\ attn\ varlen\ func()$ function to enable returning cumsum attention weights $S_{\rm cumsum}\in \mathcal{R}^{N}$, which can be transformed to $S_{\rm avg}$ via a simple transformation. Thus, our method is compatible with flash attention in practice.
>
> Based on above analysis, our method is also compatible with vllm and sglang inference frameworks which require flash attention operations, thus serving as a plug-and-play method to improve inference efficiency.
>
> 2. Performance drop on benchmarks requiring finer information.
>
> As we have discussed in the limitation section, current efficient methods are built on the idea to dynamically prune/merge input tokens to decrease computing costs to improve model efficiency, and thus easily lose key details of small objects with considerable performance drop on benchmarks requiring fine spatial information. We observe that by adopting a two-stage merging design with a novel token merging strategy, our iLLaVA could notably relieve the performance drop and outperforms previous methods on benchmarks requiring fine spatial information. As demonstrated in the following, we compare our method with previous methods upon TextVQA, ChartVQA, DocVQA and InfoVQA benchmarks to verify its effectiveness. We notice that our method can notably relieve the performance drop caused by token reduction and outperform previous methods by a large margin, which can be attributed to the advantage of our novel token merging strategy and two-stage token merging design.
>
> | Methods | TextVQA | ChartVQA | DocVQA | InfoVQA |
> | --- | --- | --- | --- | ---|
> | Qwen2.5-VL 7B (upper bound) | 84.9 | 87.3 | 95.7 | 82.6 |
> | PyramidDrop (CVPR25) | 81.1| 83.7 | 90.2 | 78.8 |
> | SparseVLM (ICML25) | 80.2| 81.4 | 88.6 | 78.1 |
> | FasterVLM (ICCV25) | 80.8| 82.5 | 89.4 | 77.5 |
> | VisionZip (CVPR25) | 81.4 | 83.2 | 90.7 | **79.3** |
> | iLLaVA (ours) | **81.9**| **84.6** | **91.6** | 79.2 |

---

> ### Author Response · Authors · 2025-11-22
> **Response for Reviewer hqY7**
>
> 3. Requirement of manual design across different models.
>
> Thanks for your question. As shown in the appendix, we first perform a manual search for the optimal insertion places in the ViT/LLM of our token merging modules based on Qwen2.5-VL. We then find that the obtained configurations can be easily transferred across different backbones like Qwen2.5-VL, InternVL2.5 and MiniCPM-V2.6 **without further modifications**. In Tab.3 of our paper, we report the performance of our method across various VLMs without modifying the insertion places, which achieve promising generalizability over different architectures. We list the results in the following for your reference. We observe that iLLaVA could show superior generalizability across different benchmarks with default configurations and notably outperform previous methods. This could be attributed to that mostly current VLMs adopt a similar architecture consisting of a vision encoder, a MLP and a LLM, which makes our model easily transfer across different VLMs. Moreover, considering most recent VLMs usually adopt the same vision encoder (e.g., SigLip) and the same VLM series (e.g., Qwen series), it’s easy for our model to adapt to different VLMs without further modifications.
>
> | Model    | MMMU | MMBench | MMStar | RealWorldQA | Avg  |
> |------------------------------|------------|-------------------|----------------|--------------|------|
> | Qwen2.5-VL           |  58.6      | 83.5              | 63.9           | 68.5         | 68.6 |
> | PyramidDrop (CVPR25)         | 57.9       | 81.6             | 60.9           | 62.7         | 65.8 |
> | SparseVLM (ICML25)           | 58.4       | 82.7              | 61.9          | 67.2       | 67.6 |
> | FasterVLM (ICCV25)           | 57.9       | 81.2              | 61.1      | 66.4         | 66.7 |
> | VisionZip (CVPR25)           | **58.3**       | 82.5             | 62.1          | 67.4         | 67.6 |
> | **iLLaVA (Ours)**            | 58.2   | **83.0**          | **62.8**           | **68.2**     | **68.1** |
> ---
> | Model                         | MMMU| MMBench | MMStar | RealWorldQA | Avg  |
> |------------------------------|------------|-------------------|----------------|--------------|------|
> | InternVL2.5 8B               | 56.0       | 84.6              | 62.8           | 70.1         | 68.4 |
> | PyramidDrop (CVPR25)         | 52.6       | 80.6              | 59.8           | 65.2         | 64.3 |
> | SparseVLM (ICML25)           | 53.1       | 81.3              | 59.2           | 65.6         | 64.8 |
> | FasterVLM (ICCV25)           | 54.2       | 82.5              | 60.2           | 66.8         | 65.9 |
> | VisionZip (CVPR25)           | 54.8       | **83.2**          | 61.3           | 68.7         | 67.0 |
> | **iLLaVA (Ours)**            | **55.4**   | 83.1              | **61.8**       | **69.4**     | **67.5** |
> ---
> | Model                         | MMMU | MMBench | MMStar | RealWorldQA | Avg  |
> |------------------------------|------------|-------------------|----------------|--------------|------|
> | MiniCPM-V2.6 8B              | 49.8       | 81.5              | 57.5           | 65.0         | 63.5 |
> | SparseVLM (ICML25)           | 47.6       | 77.8              | 54.9           | 62.1         | 60.0 |
> | PyramidDrop (CVPR25)         | 48.0       | 78.6              | 54.7           | 62.7         | 61.0 |
> | FasterVLM (ICCV25)           | 48.6       | 79.5              | 56.1           | 63.1         | 61.4 |
> | VisionZip (CVPR25)           | 48.9       | 80.2              | **56.5**       | 64.0         | 62.4 |
> | **iLLaVA (Ours)**            | **49.3**   | **81.0**          | 56.3           | **64.3**     | **62.8** |

---

### Author Response · Authors · 2025-12-01
**Overall response**

2. Additional results on benchmarks requiring finer information.

Current efficient methods dynamically prune/merge input tokens to decrease computing costs to improve model efficiency, and thus easily lose key details of small objects with considerable performance drop on benchmarks requiring fine spatial information. We observe that by adopting a two-stage merging design with a novel token merging strategy, our iLLaVA could notably relieve the performance drop and **outperforms previous methods on benchmarks requiring fine spatial information**. In the following results, we compare our method with previous methods upon TextVQA, ChartVQA, DocVQA and InfoVQA benchmarks to verify its effectiveness. Our method can notably relieve the performance drop caused by token reduction and outperform previous methods by a large margin, which can be attributed to the advantage of our novel token merging strategy and two-stage token merging design.

| Methods | TextVQA | ChartVQA | DocVQA | InfoVQA |
| --- | --- | --- | --- | ---|
| Qwen2.5-VL 7B (upper bound) | 84.9 | 87.3 | 95.7 | 82.6 |
| PyramidDrop (CVPR25) | 81.1| 83.7 | 90.2 | 78.8 |
| SparseVLM (ICML25) | 80.2| 81.4 | 88.6 | 78.1 |
| FasterVLM (ICCV25) | 80.8| 82.5 | 89.4 | 77.5 |
| VisionZip (CVPR25) | 81.4 | 83.2 | 90.7 | **79.3** |
| iLLaVA (ours) | **81.9**| **84.6** | **91.6** | 79.2 |

3. Requirement of manual design across different models.

As shown in the appendix, we first perform a manual search for the optimal insertion places in the ViT/LLM of our token merging modules based on Qwen2.5-VL. We then find that the obtained configurations can be **easily transferred across different backbones** like Qwen2.5-VL, InternVL2.5 and MiniCPM-V2.6 **without further modifications**. We list the results of our method across various VLMs without modifying the insertion places in the following for your reference. We observe that iLLaVA could show superior generalizability across different benchmarks with default configurations and notably outperform previous methods. This could be attributed to that mostly current VLMs adopt a similar architecture consisting of a vision encoder, a MLP and a LLM, which makes our model easily transfer across different VLMs. Moreover, considering most recent VLMs usually adopt the same vision encoder (e.g., SigLip) and the same VLM series (e.g., Qwen series), it’s easy for our model to adapt to different VLMs without further modifications.

| Model    | MMMU | MMBench | MMStar | RealWorldQA | Avg  |
|------------------------------|------------|-------------------|----------------|--------------|------|
| Qwen2.5-VL           |  58.6      | 83.5              | 63.9           | 68.5         | 68.6 |
| PyramidDrop (CVPR25)         | 57.9       | 81.6             | 60.9           | 62.7         | 65.8 |
| SparseVLM (ICML25)           | 58.4       | 82.7              | 61.9          | 67.2       | 67.6 |
| FasterVLM (ICCV25)           | 57.9       | 81.2              | 61.1      | 66.4         | 66.7 |
| VisionZip (CVPR25)           | **58.3**       | 82.5             | 62.1          | 67.4         | 67.6 |
| **iLLaVA (Ours)**            | 58.2   | **83.0**          | **62.8**           | **68.2**     | **68.1** |
---
| Model                         | MMMU| MMBench | MMStar | RealWorldQA | Avg  |
|------------------------------|------------|-------------------|----------------|--------------|------|
| InternVL2.5 8B               | 56.0       | 84.6              | 62.8           | 70.1         | 68.4 |
| PyramidDrop (CVPR25)         | 52.6       | 80.6              | 59.8           | 65.2         | 64.3 |
| SparseVLM (ICML25)           | 53.1       | 81.3              | 59.2           | 65.6         | 64.8 |
| FasterVLM (ICCV25)           | 54.2       | 82.5              | 60.2           | 66.8         | 65.9 |
| VisionZip (CVPR25)           | 54.8       | **83.2**          | 61.3           | 68.7         | 67.0 |
| **iLLaVA (Ours)**            | **55.4**   | 83.1              | **61.8**       | **69.4**     | **67.5** |
---
| Model                         | MMMU | MMBench | MMStar | RealWorldQA | Avg  |
|------------------------------|------------|-------------------|----------------|--------------|------|
| MiniCPM-V2.6 8B              | 49.8       | 81.5              | 57.5           | 65.0         | 63.5 |
| SparseVLM (ICML25)           | 47.6       | 77.8              | 54.9           | 62.1         | 60.0 |
| PyramidDrop (CVPR25)         | 48.0       | 78.6              | 54.7           | 62.7         | 61.0 |
| FasterVLM (ICCV25)           | 48.6       | 79.5              | 56.1           | 63.1         | 61.4 |
| VisionZip (CVPR25)           | 48.9       | 80.2              | **56.5**       | 64.0         | 62.4 |
| **iLLaVA (Ours)**            | **49.3**   | **81.0**          | 56.3           | **64.3**     | **62.8** |

---

### Author Response · Authors · 2025-12-01
**Overall response**

Many thanks for the priceless efforts and comments of the reviewers and area chair. We appreciate that most reviewers acknowledge that our work is **novel with new ideas** (Reviewer#jbVu and #MrAM), achieve **significant improvements** on widely-used benchmarks (All Reviewers) and maintain a great **accuracy-efficiency trade-off** with full ablations (Reviewer#hqY7, #AUQc and #jbVu). We hereby provide an overall response for the main concerns raised by the reviewers.

1. Differences and discussion with existing token reduction methods.

Existing token reduction methods typically **just** prune/merge redundant tokens **within the LLM stage** in VLMs, without leveraging the feature redundancy within other computation-intensive stages. In this paper, we first verify** the vision encoder consumes a significant part of computations within VLMs**, and then explore **how to better merge tokens in both stages to accelerate models**. Our contributions are two-fold: (1) We explore how to design the framework to simultaneously accelerate the vision encoder and LLM to achieve a good accuracy-efficiency trade-off. (2) We introduce a novel token merging algorithm to better recycle beneficial information from discarded tokens to preserve model performance. Results on 10+ benchmarks verify the effectiveness of our method, and comprehensive ablations confirm the efficacy of each component in our model.

We notice that a previous work (VScan) also adopts a two-stage pruning/merging design. We want to highlight the key differences between our iLLaVA and VScan as follows:

(1) **Methodology design**. VScan lacks a structured design. It simply reduces visual tokens at the penultimate layer (global scan) and 6th layer (local scan) of the visual encoder. In the LLM, it prunes a part of tokens at a single middle layer. Instead, we try to explore a **unified framework to accelerate different stages of VLMs**. We comprehensive search the token reduction positions and styles within both the LLM and the vision encoder, and introduce a new token reduction strategy applicable for all VLM stages. We try to form our method as the basis for following methods.

(2) **Token pruning/merging strategy**. In the visual encoder, VScan merges discarded tokens with preserved tokens by average pooling. In the LLM, it just prunes discarded tokens without information recycling. Instead, we design **a novel token merging strategy** which recycles beneficial information from discarded tokens to minimize information loss. Comprehensive results in our ablations show our strategy notably outperforms existing methods across several benchmarks.

(3) **Compatibility with flash attention**. VScan may not be compatible with flash attention. In the vision encoder, for modern visual encoders (e.g., SigLip) that don’t have a [CLS] token, VScan has to refer to the full attention matrix $R^{N \times N}$ to distinguish important tokens. However, as flash attention doesn’t support returning the full attention matrix, VScan may not be compatible with flash attention. Instead, our token merging process **is inherently compatible with flash attention**, which only relies on the sliced attention score $S_{avg}\in R^N$ to perform token reduction, which can be obtained via passing $return\ attn\ probs=True$ to the function $flash\ attn\ varlen\ func()$ in the forward pass of flash attention.

We conduct a series of experiments to verify the superiority of iLLaVA compared to VScan. As shown as below, we first observe iLLaVA could notably outperform VScan by around 1.0% on these four benchmarks, demonstrating its superiority brought by methodology design. We then verify the effectiveness of different components in iLLaVA by replacing each one with the design of VScan, including replacing the token reduction design in the visual encoder (denoted as visual design degradation), replacing the token reduction design in the LLM (denoted as LLM design degradation) and replacing the token merging strategy (denoted as token merging strategy degradation) to the configurations of VScan. We notice that changing each component from the design of iLLaVA to VScan would degrade the performance notably, which verifies the effectiveness of each component design in iLLaVA.

| Methods | MMMU | MMBench| MMStar | RealWorldQA|
| --- | --- | --- | --- | ---|
| iLLaVA | **56.9** | **79.7** | **60.5** | **65.3** |
| iLLaVA (Visual design degradation) | 56.1| 79.0 | 59.8 | 64.6 |
| iLLaVA (LLM design degradation) | 56.4| 79.2 |60.0 | 64.9 |
| iLLaVA (Token merging strategy degradation) | 56.3 | 79.1 | 59.8| 64.8 |
| VScan | 55.7| 78.8 | 59.5 | 64.3 |

---

### Meta-Review · Area_Chair_2ccj · 2026-01-07

**Summary:**

This paper proposes iLLaVA, a two-stage visual token merging framework for large vision–language models that reduces redundancy in both the vision encoder and the LLM by explicitly aggregating and recycling information from discarded tokens. The method is designed to be compatible with modern inference frameworks and is evaluated across a broad set of image, multi-image, and video benchmarks, demonstrating substantial gains in throughput and prefilling speed while preserving or improving accuracy under aggressive token reduction.

Reviewers raised several concerns related to `the novelty of the two-stage merging design relative to prior token pruning methods`, `potential performance degradation on fine-grained spatial tasks`, `the manual choice of merging locations and ratios`, and `practical deployment considerations`. Through detailed rebuttal and discussion, the authors clarified the distinctions from prior work, added targeted evaluations on fine-grained benchmarks, and demonstrated compatibility and robustness across multiple modern VLM backbones. Taken together, the concerns were sufficiently addressed, and the paper provides a strong and practical contribution to efficient multimodal inference.

**Reviewer Concerns:**

**1. [Partially addressed] Novelty relative to prior two-stage token pruning / merging methods**
(by: AUQc, jbVu, MrAM)

Multiple reviewers questioned the novelty of iLLaVA, noting that reducing tokens at both the vision encoder and LLM stages, as well as merging discarded tokens, has been explored in prior work such as VScan. The concern was that the contribution might rely primarily on design refinements (e.g., progressive merging, layer placement) rather than introducing a fundamentally new token reduction paradigm.

The rebuttal provided a detailed comparison with prior methods, clarifying distinctions in merging strategy (explicit information recycling rather than hard pruning or simple pooling), progressive reduction design, and inference compatibility. While these clarifications and ablations demonstrate that the proposed design choices lead to consistent empirical gains, the differences remain incremental in nature. As such, the novelty concern is partially addressed.

**2. [Addressed] Performance degradation on fine-grained, spatially sensitive benchmarks**
(by: hqY7)

Reviewers expressed concern that aggressive token merging could harm tasks requiring fine spatial details (e.g., DocVQA, ChartVQA, OCR-style benchmarks).

The authors added comprehensive evaluations on multiple fine-grained benchmarks (TextVQA, ChartVQA, DocVQA, InfoVQA), demonstrating that iLLaVA consistently outperforms existing efficient baselines under comparable token budgets. This concern was addressed.

**3. [Addressed] Compatibility with FlashAttention and mainstream inference frameworks**
(by: hqY7)

One reviewer questioned whether the method is compatible with FlashAttention and deployment frameworks such as vLLM and SGLang.

The authors clarified that iLLaVA relies only on sliced or aggregated attention statistics compatible with FlashAttention-based inference and explicitly demonstrated integration with common inference frameworks. This concern was addressed.

**4. [Partially addressed] Manual configuration of token reduction ratios and insertion layers**
(by: hqY7, jbVu)

Reviewers noted that reduction ratios and insertion positions are manually specified, raising concerns about tuning cost and portability.

The rebuttal showed that configurations tuned on one backbone transfer well to several others without further tuning. While an automatic strategy is not provided, the demonstrated robustness partially alleviates the concern.

**5. [Addressed] Insufficient explanation and visualization of the token merging process**
(by: jbVu, MrAM)

Some reviewers requested clearer explanations and visualizations of the token merging and recycling process.

The authors expanded the methodological explanation, added detailed visualizations, and provided targeted ablations, which sufficiently improved clarity.

**Reviewer Scores:**

**Reviewer hqY7 (4 -> 6)**

Reviewer hqY7 focused on fine-grained task performance, deployment compatibility, and manual configuration choices. The rebuttal convincingly addressed performance and compatibility issues and partially alleviated concerns about portability, which would likely strengthen the reviewer’s assessment.

**Reviewer AUQc (4 -> 5)**

Reviewer AUQc raised substantive concerns about novelty relative to prior two-stage pruning and merging methods. While the rebuttal did not fully eliminate the sense of incremental contribution, the detailed comparisons and additional ablations clarified the practical distinctions and empirical benefits of the proposed design. From an AC perspective, these clarifications would likely lead to a modest upward revision.

**Reviewer jbVu (6 -> 7)**

Reviewer jbVu was generally positive but raised concerns about novelty, fine-grained performance, and clarity of the merging process. The added fine-grained benchmark results and improved explanations addressed most of these concerns, supporting a moderate increase in score.

**Reviewer MrAM (6 → 7)**

Reviewer MrAM appreciated the efficiency gains but requested clearer intuition and visualization of the token merging process. The expanded explanations and visualizations improved clarity and interpretability, justifying a modest upward adjustment.

---

### Decision · Program_Chairs · 2026-01-26

Accept (Poster)